# Investigation of a Sparse Autoencoder-Based Feature Transfer Learning Framework for Hydrogen Monitoring Using Microfluidic Olfaction Detectors

**DOI:** 10.3390/s22207696

**Published:** 2022-10-11

**Authors:** Hamed Mirzaei, Milad Ramezankhani, Emily Earl, Nishat Tasnim, Abbas S. Milani, Mina Hoorfar

**Affiliations:** 1School of Engineering, University of British Columbia Okanagan Campus, Kelowna, BC V1V 1V7, Canada; 2Department of Mechanical Engineering, University of Victoria, Victoria, BC V8P 5C2, Canada

**Keywords:** hydrogen detection, HENG, transfer learning, sparse autoencoder, microfluidic gas sensor

## Abstract

Alternative fuel sources, such as hydrogen-enriched natural gas (HENG), are highly sought after by governments globally for lowering carbon emissions. Consequently, the recognition of hydrogen as a valuable zero-emission energy carrier has increased, resulting in many countries attempting to enrich natural gas with hydrogen; however, there are rising concerns over the safe use, storage, and transport of H2 due to its characteristics such as flammability, combustion, and explosivity at low concentrations (4 vol%), requiring highly sensitive and selective sensors for safety monitoring. Microfluidic-based metal–oxide–semiconducting (MOS) gas sensors are strong tools for detecting lower levels of natural gas elements; however, their working mechanism results in a lack of real-time analysis techniques to identify the exact concentration of the present gases. Current advanced machine learning models, such as deep learning, require large datasets for training. Moreover, such models perform poorly in data distribution shifts such as instrumental variation. To address this problem, we proposed a Sparse Autoencoder-based Transfer Learning (SAE-TL) framework for estimating the hydrogen gas concentration in HENG mixtures using limited datasets from a 3D printed microfluidic detector coupled with two commercial MOS sensors. Our framework detects concentrations of simulated HENG based on time-series data collected from a cost-effective microfluidic-based detector. This modular gas detector houses metal–oxide–semiconducting (MOS) gas sensors in a microchannel with coated walls, which provides selectivity based on the diffusion pace of different gases. We achieve a dominant performance with the SAE-TL framework compared to typical ML models (94% R-squared). The framework is implementable in real-world applications for fast adaptation of the predictive models to new types of MOS sensor responses.

## 1. Introduction

Blending hydrogen into the existing natural gas pipeline network has to reduce greenhouse gas emissions from natural gas production, distribution, and consumption fields [1]. Reports show that conventional NG pipelines can carry a blend of natural gas and hydrogen of up to approximately 15% by volume of hydrogen while only requiring modest modifications to the pipeline [2]. While hydrogen is a valuable and clean alternative for carbon-based fuels, its unique physicochemical properties make it a highly permeable and explosive gas that requires precise monitoring. Hydrogen gas is odorless, colorless, and non-toxic. Therefore, it cannot be sensed by the human olfactory system. Due to this, enhanced safety measures must be put in place to monitor its presence and concentration in various environments. These sensors are required to monitor leakage of hydrogen, and measure hydrogen concentration in all processes involving production, transportation, and storage. The gold standard methods used to analyze gaseous molecules are gas chromatography-based and mass spectrometry-based technologies [3], but these platforms are relatively large, expensive, and somewhat slow.

On the other hand, hydrogen sensors are cost-effective, small, easy to operate, and can be highly sensitive and selective. Hydrogen sensors have been produced for several decades. While several commercialized hydrogen sensors are already available, the efforts to improve them for continuous and precise real-time monitoring for the future hydrogen-based economy are ongoing. These efforts focus on improving the signal’s duration, the sensors’ selectivity and sensitivity, and the miniaturization of the sensors.

Numerous technologies have been developed to detect hydrogen, such as optical sensors [4,5], electrochemical sensor fields [6,7], catalytic sensors [8,9], work function sensors [10,11], and resistance-based sensors [12,13]. Among these devices, resistance-based sensors, specifically metal–oxide–semiconductor (MOS) sensors, have gained significant attention in gas detection due to their relatively small size, low cost, ease of operation, and high sensitivity [14]. As the sensing mechanism in a MOS sensor is based on the adsorption and the consumption of the oxygen molecules (by reducing gas in *p*-type semiconductors and vice versa in *n*-type semiconductors) on the surface of the sensing layer, these sensors can detect a wide range of gaseous elements such as volatile organic compounds, hydrocarbons, and combustible gases, including hydrogen. However, this wide detection range limits specificity and sensitivity for certain analytes. To overcome this issue, a 3D-printed microfluidic channel can be used to delay the diffusion of the target gas onto the surface of the sensing layer. Using novel coating compositions and surface treatments on the channels’ inner walls, MOS–microchannel platforms can noticeably enhance the selectivity of MOS sensors to various target gases [15].

The resulting response curves generated from MOS-based microfluidic detectors must be processed using statistical analysis, namely feature extraction, to more accurately estimate the hydrogen concentration. Many feature extraction methods implemented in the literature attempt to generate a condensed representation (i.e., dimensionality reduction) of the signals produced by sensors [16]. Given the limited available gas sensor data, the high-dimensional input feature space of the response curves must be translated into a low-dimensional feature representation to prevent the “curse of dimensionality” causing poor generalization performance (i.e., overfitting) in the Machine Learning (ML) model [17]. In one common approach, the desired features are extracted by the expert user based on domain knowledge, experience, and the geometric characteristics of the response curve [18]. The most commonly extracted features include the maximum response value, the area under the curve, and the sensor response’s rising and falling slopes/time [17]. The drawback of this method is that the feature extraction process needs to be performed manually, as the meaningful features for each gas/sensor response curve can vary. It is also likely that some informative features may be overlooked and excluded from data analysis.

Unsupervised learning methods are powerful dimensionality reduction techniques used to learn and extract meaningful information from sensor responses [18]. Principal component instance and principal component analysis (PCA) are considered liable unsupervised techniques, widely used for reducing the dimensionality of the input feature space before being fed into the ML model. The principal components’ direction in PCA is calculated to maximize the described variance of the original dataset [19]. As such, PCA is a popular choice for achieving low-level representations of data. Yin and Tian [20] utilized PCA to reduce the dimensions of gas sensor data for classifying Chinese drinks. The advantage of PCA over the above feature extraction methods is that it automatically learns the most important elements in the dataset while being robust to the presence of correlated input variables. However, PCA fails to extract useful features in highly non-linear feature spaces as it is limited to linear projections [17].

Similarly, other traditional dimensionality reduction approaches, such as manifold learning methods, fall short in learning high-level abstractions [21]. Deep learning models offer more flexibility than conventional ML models and non-linear data structures. In other words, due to the non-linearity in their activation functions, they are capable of learning complex operations and highly non-linear feature representations. Autoencoders (AEs) are deep unsupervised learning methods designed for learning dense latent features and dimensionality reduction [22]. 

Once the latent features have been determined, the pre-processed data usually is fed to a classification/regression model. In gas detection, efforts have been made to improve the performance of electronic nose (e-nose) devices by incorporating deep learning (DL) models. Zhao et al. [23] utilized a deep learning model to differentiate various Chinese liqueurs based on the responses from electronic noses. Ma et al. [24] used the images produced by a sensor array to train a Convolutional Neural Network (CNN) to detect unknown gases. The result indicates that CNN’s performance is superior to traditional ML approaches. Despite their outstanding performance, deep learning models require large datasets to be trained properly. In the absence of a sufficient amount of data, deep learning models often overfit the training dataset (due to many trainable parameters), thus generalizing poorly on new unseen instances.

One way to mitigate the performance-declining effect of limited data is to implement Transfer Learning (TL) [25]. TL leverages knowledge from related domains and incorporates it into the learning process for tasks of interest with limited data. This significantly reduces the dependency complex models, such as DL-based methods, have on large datasets while improving the model’s generalization performance [25]. Yan and Zhang [26] developed a sample-based transfer learning using Ridge and Logistic regressions for tackling the domain distribution shifts caused by sensor drifts and instrumental variation. To address the sensor drift, the proposed framework by Yi et al. [27] initially minimizes the domains’ distribution inconsistency by minimizing the Maximum Mean Discrepancy (MMD), followed by an adaptive Extreme Machine Learning (EML) classifier.

Although some research has been performed on mitigating the effect of instrumental variation and sensor drift on ML performance for various types of gases and e-noses, investigating and addressing such a phenomenon for hydrogen-enriched natural gas (HENG) mixtures remains unexplored. In this paper, a Sparse Autoencoder-based Transfer Learning (SAE-TL) is developed to address the issue of the sensor response shift in an in-house 3D printed microfluidic channel coupled with two commercial MOS sensors for estimating the hydrogen gas concentration in HENG mixtures using limited datasets. Using source data, the framework first learns a low-level representation of the response curves by SAE through an unsupervised learning procedure. Then, the learned words are used in the TL portion of the framework to handle the domain shifts and learn a reliable regressor for a new sensor model with very few available data. 

## 2. SAE-TL Framework

The schematic of the proposed framework is illustrated in Figure 1, and the details are elaborated in the following sub-sections.

### 2.1. Sparse Autoencoder (SAE)

An AE is comprised of an encoder and a decoder. The encoder, which consists of the input and hidden layers, receives high-dimensional raw data and transforms it into a low-level latent representation, while the decoder (the hidden and output layers) is responsible for reconstructing the original data from the hidden layer output, i.e., condensed representation. The hidden layer typically contains fewer neurons than the input and output layers, so a denser representation of the raw data can be obtained [18]. AEs learn to minimize the distance between the raw signal x and its reconstructed version $\hat{x}$ by minimizing the reconstruction loss,
(1)$$J_{\mathrm{AE}}\left(\theta \right)=\sum_{i=1}^{n}L\left(x_{i},\;\hat{x_{i}}\right)$$
where $\theta \;=\;\left\{W,\;b,\;W^{\prime },b^{\prime }\right\}$ denotes the weights and biases of the encoder and decoder, respectively, and n is the dataset size.

Due to the fact that AEs are designed to identify meaningful features from raw data and reconstruct the original data at the output layer, they are prone to copying the raw data from the input layer to the output layer without extracting any useful features [22]. To avoid this, one approach is to monitor the activation of each neuron in the hidden layer and penalize the units that have an activation higher than the specified threshold using a sparsity loss term. As a result, the SAE learns useful representations while attempting to avoid large activations in the hidden layer. Kullback–Leibler (KL) divergence is used as the sparsity loss and can be represented as (2).
(2)$$\mathrm{KL}\left(q||p_{k}\right)=q\log\frac{q}{p_{k}}-\left(1-q\right)\log\frac{1-q}{1-p_{k}}$$

It measures the distance between the predefined sparse parameter q (usually set to be a small value) and the average activation of each hidden unit:(3)$$p_{k}=\frac{1}{n}\sum_{i=1}^{n}\left[s_{f}\left(b_{k}+W_{k}x_{i}\right)\right]$$
where $s_{f}$ denotes the hidden layers activation function. Thus, the cost function of the SAE can be written by combining the AE cost function and the KL divergence measure as follows:(4)$$J_{\mathrm{SAE}}\left(\theta \right)=\sum_{i=1}^{n}L\left(x_{i},\;\hat{x_{i}}\right)+\mathrm{KL}\left(q||p_{k}\right)$$

### 2.2. Transfer Learning

Deep learning approaches have been shown to be a powerful tool in deriving important features from the gas sensor and e-nose response curves [28,29]. Despite the promising results, deep learning approaches require large amounts of data for training and are usually ill-equipped to handle slight shifts in data distribution. For example, implementing a new sensor with similar but different specifications will cause the model to be unable to perform accurate prediction tasks. One solution to alleviate this dependency on large datasets is to utilize TL [28]. In a TL framework, the knowledge gained from a model that is trained on sufficient data (the source) is used for improving the performance of a model that is trained on the task of interest that has limited data (the target) [30]. When considering HENG mixtures detection tasks, historical experimental data has been generated from a specific type of sensor. An ML model trained on such data may be ineffective when used against a new sensor response (i.e., domain shift), as it will exhibit different behavior. However, it is also known that the response curves of gas sensors share considerable similarities, suggesting that a model trained on historical data can still be useful as it conveys the learned knowledge about the general behavior of the response curve. This knowledge can significantly decrease the data required for learning low-level available feature representations if transferred to the target model via TL. In Neural Networks (NN), TL can be incorporated by transferring the fine-tuned weights from the source model to the target model. Using this approach, the target network requires only a few gradient steps using the limited available data to reach optimal performance [31].

For an input space *X* and a label space *Y*, the ML objective predictive function f: X→Y can be broken down into two corresponding functions: f=h∘g. While limitg: X→Z maps the input space to a latent low-dimensional feature space, Z, the embedding function limith: Z→Y is a regressor (a predictive function) that predicts the output y using the feature space. It has been shown that, when training NNs on different but related datasets, the first layers of the networks learn similar low-level feature representations. This general behavior is seen in all networks regardless of the model specifications (e.g., the cost function and the input data structure) [32]. On the contrary, the final layers of networks developed using related data sets exhibit completely different behavior. Towards their final layers, each network becomes more specific to the task they are being trained on. This phenomenon is observed in networks trained on various types of data (e.g., tabular, image, and text) [33].

For an SAE to learn low-level feature representations of gas sensors’ response curves, it requires training the SAE using a large dataset. In cases where data is limited, i.e., a new sensor is implemented (instrumental variation), training the SAE and the classifier is not immediately possible. One infeasible remedy is to generate a large dataset using the new sensor, which causes high temporal and financial costs. A more efficient solution is to implement TL. In the proposed TL framework, low-level feature representations of the source sensor’s response curves are initially learned using SAE.

Once learned, the SAE’s encoder ($g_{s}$) is connected to a randomly initialized multilayer perceptron (MLP) (regressor, $h_{s}$) to form the source model. In a supervised regime, the labeled source data is used to train the source model in two steps. The initial weights of the target regressor produce large error gradients during the back-propagation step, which can destructively modify the learned weights of the transferred encoder. Initially, the encoder’s weights are fixed (frozen) to avoid large error gradients. After a few epochs, as the regressor’s weights are stable and the error gradient shrinks, the whole model becomes unfrozen and is trained until it reaches its optimal state. Once the source network is developed ($f_{s}=h_{s}\circ g_{s}$), it is transferred to the target model (Figure 1). Next, the target model needs to be fine-tuned using the limited available target data. This step is necessary for the model to learn more specific features/trends of the target task, boosting its overall performance. However, training the whole network requires a large dataset. Therefore, the target encoder ($h_{T}$), transferred from the source model, is frozen during the fine-tuning process. This ensures that the knowledge learned from the general layers of the source model is transferred and remains intact (i.e., avoiding forgetting the learned knowledge). The target regressor ($h_{T}$), which contains more task-specific layers, will be fine-tuned with the target data. This allows the SAE to bypass the data-intensive procedure of learning low-level features, resulting in a network that requires only a small dataset to be re-calibrated to the desired task [34].

## 3. Experimental Verification

### 3.1. Case Study: Hydrogen Gas Detection Using a Microfluidic Detector

Hydrogen is a light and odorless gas; it is challenging to determine the concentration in the mixture of HENG in real time using commercial sensors [35]. The use of a microfluidic channel for the detection of gases based on adsorption/desorption phenomena provides the apparatus with enough delay in the diffusion time and, consequently, in the response time to differentiate between various gases [15,28,36]. The schematic diagram of the experimental sensing apparatus is illustrated in Figure 2. The sensor platform consists of a MOS sensor (Figaro TGS2610 or Figaro TGS2611), explosion-proof valves, a mixing chamber, mass flow controllers (MFCs), an automated sensor housing carrier, electronic components, and tubing. By calculating the chamber size and the passing flow rate, this automatic setup supplies the mixing chamber with the desired concentration of a single gas or mixture of target analytes. Exhaust valves have been used to regulate residual gas pressure, which will be trapped between the MFCs and the chamber inlet valves. Before being exposed to the sensor, the gas mixture rests in the mixing chamber to ensure a homogenous state. During this period, the data collection process begins to generate a baseline signal for the sensor. After a homogenous state is attained, the microchannel inlet is exposed to the gas chamber by opening a magnetic valve (demonstrated in Figure 3). The gas mixture is then diffused through the microfluidic channel and reaches the sensing layer. Unique raw data for each experiment will be automatically collected in a RaspberryPi 4 and progressed in our TL framework. Due to the specific diffusion times of different gases and the adsorption/desorption phenomena that occur on the channel walls, individual responses are obtained for every gas mixture [15], providing the sensing platform with improved selectivity. Because of a large number of tests and to avoid a different MOS sensor drift impact [37], Figaro TGS2610 and TGS2611 sensors were switched after every five experiments. In order to not only make changing sensors feasible but also to ensure that the microchannel effect on gas responses was kept constant, a modular 3D-printed housing system was designed (Figure 4), allowing for different sensors to be mounted at the end of the microfluidic channel.

Based on a previous study to maximize the selectivity between VOCs, the channel length was set to 30 mm, with a width of 3 mm and a height of 1 mm. As reported in previous and parallel works done by our research group, these microchannel-MOS gas detectors can provide specific patterns for various gases and are used in several gas detection applications, such as wine identification, natural gas detection, hydrogen sulfide monitoring in sewer systems, and differentiating between VOCs [15,28,36,38].

### 3.2. Sensor Characteristics

The utilized sensors, TGS-2610 and TGS-2611, are semiconductor-based, and their output is raw voltages from the sensing layer before, during, and after exposure to a target gas, as illustrated in Figure 3. Conventional HENG monitoring methods have proven to have a 90–270 s response time (including the recovery time) [39,40,41,42,43]. In this work, taking advantage of a microfluidic channel working as a delaying agent, exponential shape responses are generated from the interaction between the hydrogen-enriched natural gas and semiconductor-based and commercially available gas sensors. As demonstrated in Table 1, the response time for this platform is as low as 150 s. This relatively short and real-time response ensures a fast data collection procedure as part of the training of the proposed learning framework. Table 1 summarizes and compares key characteristics of available industrial and pilot HENG monitoring methods.

### 3.3. Gas Mixture

Binary mixtures of methane and hydrogen were used for this study, as they are the two main components of HENG. The desired concentration of the HENG mixture was mixed from compressed gas tanks of 99.9% methane and hydrogen, respectively (Praxair Canada, Mississauga, ON, Canada). Due to safety considerations and reported optimal percentages of hydrogen in HENG, HENG mixtures of up to 10% (*v*/*v*) hydrogen were prepared. As illustrated in Figure 4, hydrogen and methane pressures were regulated to MFC’s operating pressure to obtain a homogenous mixture. A specific amount of each gas was added to the gas chamber. Various methane and hydrogen mixture concentrations (20–1000 ppm) were obtained using the automated experimental setup to generate data for validating the proposed framework. In particular, 120 data points were generated from the source sensor (TGS2611), while 102 samples were collected from the target sensor (TGS2610). Each experiment was repeated three times to ensure the repeatability of the obtained responses.

### 3.4. SAE-TL Experimental Design

In SAE, symmetrical architecture concerning the latent representation layer (central hidden layer) is used, where the decoder mirrors the number of layers and neurons of the encoder. The main hidden layer contains 20 neurons, yielding a 20-dimensional representation of the raw data. The top-performing SAE encoder (and decoder) consists of two hidden layers with [500, 200] neurons. All coatings (except the decoder’s output layer) are equipped with the RELU activation function. Mean Squared Error (MSE) is chosen as the loss function (SAE reconstruction loss), and an Adam optimizer with a learning rate of 0.001 is implemented. Early stopping with the patience of 50 is used as the regularization term. The sparsity threshold of 10e-5 is selected for the latent representation layer. The TL portion of the framework contains two identical NNs, namely, the source and the target. Each network has an encoder with the same architecture as the SAE’s encoder. The network’s regressor is a four-layer NN with [10, 8, 6, 1] neurons. The RELU activation function and Adam optimizer with a learning rate of 0.001 were used for both networks. The whole framework is developed and trained using the Keras library in Python. For preparing the source network, the source data set is divided into training, validation, and testing (70%, 10%, 20%). For the target network, however, to mimic the available limited data in real-world scenarios, only 20% of data is allocated for training. The rest is kept to evaluate the generalization performance.

## 4. Results and Discussion

### 4.1. SAE Performance Evaluation

The performance of the SAE on the reconstruction of the sensor responses is compared with regular (vanilla) AE and PCA, and the results are summarized in Figure 3. For PCA, the first five principal components are used to reconstruct the original response. For SAE and AE, the decoder was used to map the low-level feature space to the actual distance (see Figure 1). The reconstructed responses of the SAE closely follow the ground truth curves. At the same time, both AE and PCA fall short in capturing the general behavior of the sensor response. The dominant performance of the SAE can be explained by its capability to capture non-linear representations (unlike PCA) and the effect of sparsity (regularization), which makes the model more robust against overfitting than AE.

The effect of the number of hidden layers (i.e., encoder layers + latent representation layer + decoder layers) of the AE and SAE on their reconstruction performance is also studied. Adding more hidden layers can give the model more flexibility to learn nonlinearity in the feature space better. On the other hand, it increases the number of hyperparameters, which may lead to overfitting during the training. Table 2 summarizes the model MSE loss when trained with 1–9 hidden layers. Though having a minor effect, increasing the number of hidden layers seems to reduce the reconstruction MSE reaching its minimum at five hidden layers. Adding more layers does not further reduce the models’ loss, which can indicate overfitting due to the large number of parameters added by each layer. For the rest of the paper, results of SAE and AE with five hidden layers are reported.

### 4.2. TL Performance Evaluation

To evaluate the performance of the proposed TL method, it is compared with conventional state-of-the-art ML approaches. For the dimensionality reduction, PCA was performed on two datasets: (1) the limited target data and (2) the source and target data combined. The reduced representations, i.e., top five principal components, were then passed to a regression model, namely, a Support Vector Machine (SVM) [44], a Random Forest (RF) [45], and an XGBoost [46]. Mean Absolute Error (MAE) and R-squared of each model against the target test data are measured and summarized in Table 3.

SAE-TL demonstrates a clear dominant performance in estimating the hydrogen concentration with an MAE of 89.24. Combining the source and target datasets for the PCA input slightly improved the conventional models’ performance. This is expected, as PCA, an unsupervised learning model, can benefit from a larger sample size to output a more robust representation. The AE-TL outperforms the PCA-based models, though it falls behind the SAE-TL due to its higher reconstruction loss (i.e., generating less-accurate representations). Finally, the NN directly trained on raw data (i.e., no dimensionality reduction) exhibits similar performance to that of the PCA-based models. The network is prone to overfitting due to the limited dataset, which reduces the model’s generalization performance.

## 5. Conclusions

This paper presents a novel TL framework for predicting the HENG mixture concentration with limited data extracted from a microfluidic-based gas detector. Our in-house 3D-printed microfluidic channel works as a delay agent and provides a unique fingerprint response [15]. The proposed framework consists of an SAE for extracting useful low-dimensional representations from the high-dimensional raw detector response data, and a TL method for transferring the learned knowledge from historical data towards a model based on data from a new detector. The response curve data from two sensors have been collected to evaluate the proposed SAE-TL method’s performance. The results clearly outline the proposed method’s dominant performance compared with conventional ML models. Namely, the SAE outperformed AE and PCA in learning more useful features for dimensionality reduction. The implementation of TL drastically reduced the need for large amounts of data for training deep neural networks. The proposed learning framework can effectively work with very limited data to learn the mapping between the sensor response and the gas concentration via the TL module in the framework. Conventional learning models require large datasets (especially since the input space, namely, sensor voltage response, is high-dimensional) to be fully trained and yield high accuracy. In contrast, our framework can enable a fast yet much less data-intensive training procedure to produce high predictive performance. In addition, many of the previous works have implemented conventional (and linear) dimensionality reduction methods such as PCA as part of the learning process; though effective, we have shown that by using a nonlinear counterpart (i.e. sparse autoencoder), a more informative low-dimensional representation of the input space can be achieved, directly leading to better predictive accuracy. Future studies are required to further validate the proposed framework’s generalizability, and response data from other types of sensors can be collected and examined. Moreover, probabilistic models such as Gaussian processes [47] or Bayesian deep learning [48] can be utilized to quantify the model uncertainty.

## Figures and Tables

**Figure 1 sensors-22-07696-f001:**
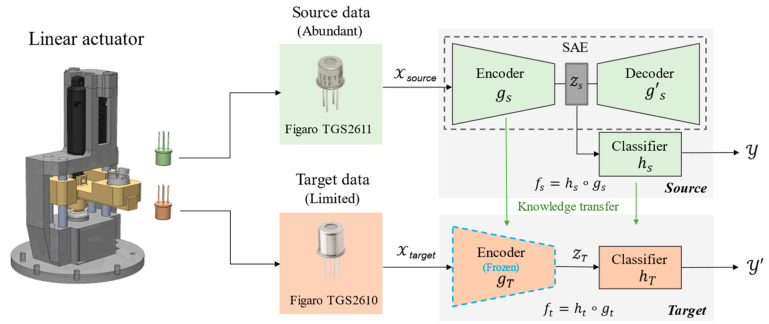
Schematic of the proposed SAE-TL framework.

**Figure 2 sensors-22-07696-f002:**
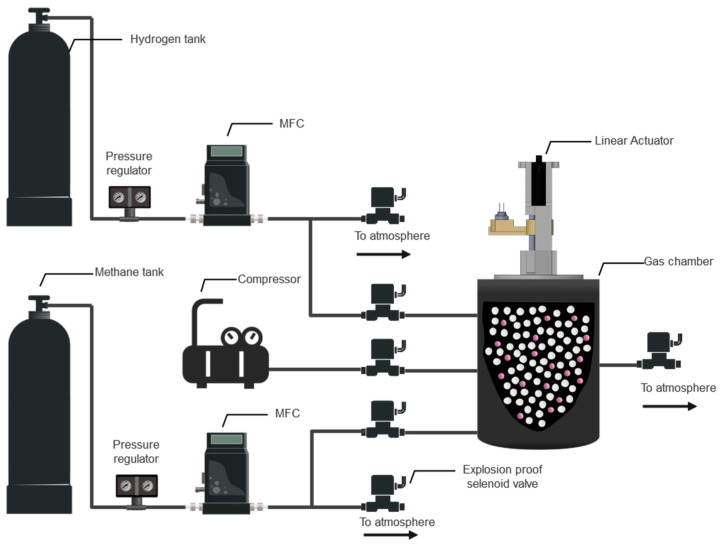
The schematic diagram of the automated experimental setup, designed and built for sample generation and data collection.

**Figure 3 sensors-22-07696-f003:**
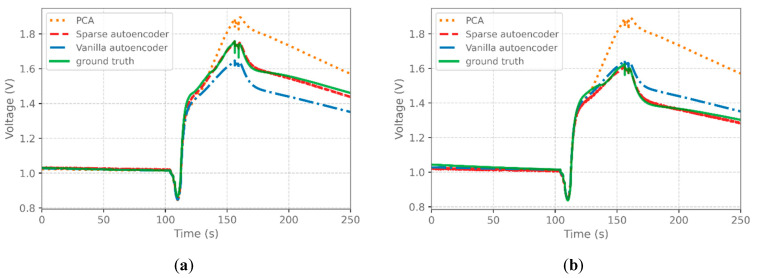
Comparison of reconstruction accuracy of SAE, AE, and PCA for gas mixture samples with (**a**) 810 (90) ppm and (**b**) 90 (10) ppm Hydrogen (Methane).

**Figure 4 sensors-22-07696-f004:**
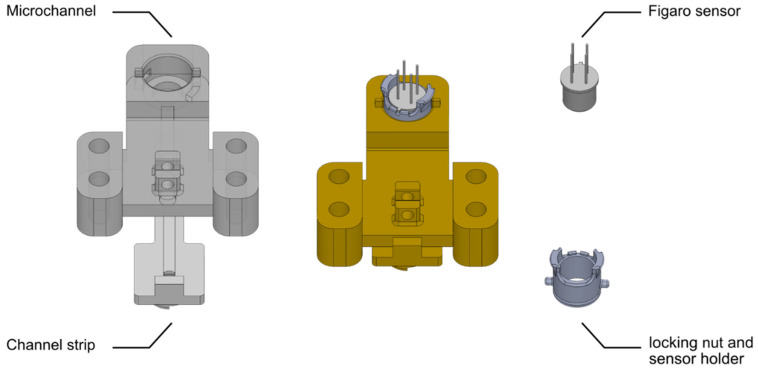
Parts and assembly of the 3D printed microchannel and the modular housing.

**Table 1 sensors-22-07696-t001:** Comparison of select hydrogen and natural gas mixture monitoring platforms by key characteristics.

Ref	Sensor Type	Limit of Detection	Power Consumption	Time from Collection to Results
[39]	Carbon nanotube/SiO_2_	Not reported	Not reported	1132 s
[40]	Calorimetric Pd/θ-Al_2_O_3_	200 ppm	0.12 W	Not reported
[41]	Au/SnO_2_, Pt/Cu/SnO_2_	500 ppm	200 mW	180 s
[42]	Pd/Au optical sensor	987 ppm	Not reported	90 s
[43]	Semiconductor, Catalytic, electrochemical	200 ppm	Not reported	190–270 s
This Work	Semiconductor	89 ppm	200 mW	150 s

**Table 2 sensors-22-07696-t002:** Effect of number of hidden layers on the performance (MSE) of SAE and AE.

	Number of Hidden Layers				
	1	3	5	7	9
**SAE**	0.00035 ± 0.00001	0.00031 ± 0.00002	**0.00024 ± 0.00001**	0.00034 ± 0.00002	0.00039 ± 0.0001
**AE**	0.0017 ± 0.00002	0.001 ± 0.00001	**0.0007 ± 0.00001**	0.008 ± 0.0001	0.0008 ± 0.0001

**Table 3 sensors-22-07696-t003:** Generalization performance of the proposed TL method on test sensor data.

Encoder	Regressor	MAE (ppm)	R-Squared
SAE-TL	MLP	**89.24**	**0.94**
AE-TL	MLP	99.74	0.89
PCA (Source)	XGBoost	172.77	0.82
	RF	206.53	0.67
	SVM	202.47	0.63
PCA (Source + Target)	XGBoost	172.77	0.82
	RF	176.82	0.73
	SVM	109.88	0.87
NN		121.25	0.84

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
