# Peer review of "Investigation of a Sparse Autoencoder-Based Feature Transfer Learning Framework for Hydrogen Monitoring Using Microfluidic Olfaction Detectors"

_sensors, 2022, doi:10.3390/s22207696_

Round 1
Reviewer 1 Report
The manuscript (sensors-1891997), Investigation on A Sparse Autoencoder-Based Feature Transfer Learning Framework for Hydrogen Monitoring using Microfluidic Olfaction Detectors, shows interesting results for microfluidic olfaction detectors. Authors presented quite comprehensive analysis in the manuscript. Some minor comments would like to provide here for authors' reference, hope it can improve manuscript for further publication, shown as following -
1. Authors should provide a simple benchmark Table to compare the current work and other research works, e.g., accuracy, power requirement, area, speed etc.... This would be quite helpful for this referee and potential readers to get further insight of the research impact and contribution in this work as compared to other works.
2. Except for accuracy improvement, did authors also find out the speed improvement in this work by using the authors' methods? How many percentage improvement authors observed as compared to conventional ways?
3. Title should be "Investigation..."
Due to the above comments, this referee would like to put the manuscript status as "Minor Revision" in the current phase.
Author Response
Dear reviewer,
The authors would like to thank you for your valuable comment sincerely. Regarding your constructive feedback, the manuscript was carefully revised, and changes have been applied in the revised text. Please find the authors' response to the comment also enclosed below. Also, please see the revised version of the paper attached as a document.
Kind regards,
- Authors should provide a simple benchmark Table to compare the current work and other research works, e.g., accuracy, power requirement, area, speed, etc. This would be quite helpful for this referee and potential readers to get further insight into the research impact and contribution in this work compared to other jobs.
Thank you very much for this suggestion. The literature on applying unsupervised AI methods in hydrogen-enriched natural gas (HENG) detection is sparse. Table 2 in the paper provides a generic comparison between the proposed framework and other common learning methods to evaluate its effectiveness (i.e., generalization performance). As suggested by the reviewer, a benchmark table is added (Table 3) in the new version of the paper to compare this study with other related works in terms of limit of detection, power consumption, and time.
- Except for accuracy improvement, did the authors also find speed improvement in this work using the authors' methods? How many percentage improvements do authors observe as compared to conventional ways?
Conventional hydrogen monitoring methods have proven to have a 90-270 s response time (including the recovery time). In this work, taking advantage of a microfluidic channel working as a delaying agent, responses are generated from the interaction between the hydrogen-enriched natural gas and semiconductor-based and commercially available gas sensors. As demonstrated in the benchmark table, the response time for this platform is as low as 150 s. This enables a relatively fast data collection procedure as part of the training of the proposed learning framework. Additionally, once the model is trained, the prediction of HENG concentration can be obtained very fast (less than a second), which further reduces the total amount of time required for this AI-assisted sensory system to report concentrations.
- The title should be "Investigation..."
As suggested, the title is now changed to "Investigation of A Sparse Autoencoder-Based Feature Transfer Learning Framework for Hydrogen Monitoring using Microfluidic Olfaction Detectors."
Reviewer 2 Report
This manuscript describes a hydrogen monitoning sensor based on a Sparse Autoencoder-based Transfer Learning (SAE-TL). The topic is interesting and some preliminary results have been obtained. But some points needs to be addressed.
1. The format and English should be checked carefully to avoid any errors such as the using of semicolon.
2. It is not clear the novelty and advantages of this system compared with current available system for hydrogen monitoring.
3. How about the specificity, sensitivity, and stability of this system? Since the gas sensors used is commercial available, the main improvement on the performance?
4. Since the gas sensors used in this study are commercial available, where is the original sources of the main improvement on the performance?
5. In a complex environment, could this sysytem work with high performance?
Author Response
Dear reviewer,
The authors would like to thank you for your valuable comment sincerely. Regarding your constructive feedback, the manuscript was carefully revised, and changes have been applied in the revised text. Please find the authors’ response to the comment also enclosed below. Also, please see a revised version of the manuscript attached as a document.
Kind regards,
- The format and English should be checked carefully to avoid errors such as the use of semicolon.
Thanks to the reviewer for pointing this out. We humbly accept that the English in this work could be improved, and for the new version, we went through the manuscript again and improved the language.
- It is not clear the novelty and advantages of this system compared with current available system for hydrogen monitoring.
Aside from the above features and advantages on the gas detection side, the proposed machine learning framework can effectively work with very limited data to learn the mapping between the sensor response and the gas concentration via the transfer learning module in the framework. Conventional learning models require large datasets (especially since the input space, namely, sensor voltage response, is high dimensional) to be fully trained and yield high accuracy, whereas our framework can enable a fast yet much less data-intensive training procedure to produce high predictive performance. Besides, many of the previous works have implemented conventional (and linear) dimensionality reduction methods such as PCA as part of the learning process, Though effective, we have shown that by using a nonlinear counterpart (i.e., sparse autoencoder), a more informative low-dimensional representation of the input space can be achieved, directly leading to a better predictive accuracy as illustrated in Figure 4 and Table 2. We included the above explanation to further clarify the novelty of our proposed framework.
- How about the specificity, sensitivity, and stability of this system? Since the gas sensors used is commercially available, the main improvement in the performance?
Thank you very much for pointing this out. Indeed we accept that sensitivity, selectivity, and stability of gas sensing platforms are integral criteria for evaluation of their performance. In the new version of the paper, we mentioned that this microfluidic-based gas detector’s sensitivity, selectivity, and stability have been investigated and proven in multiple previous and parallel researches for a wide spectrum of gases spanning from hydrocarbon natural gas to volatile organic compounds and hazardous gases such hydrogen sulfide in our research group [1-5]. We’d like to humbly point out that the novelty of this work is on AI integration and making the training fast and less data-intensive.
- Since the gas sensors used in this study are commercially available, where is the original sources of the main improvement on the performance?
The proposed learning framework provides near real-time analysis capabilities to identify the exact concentration of the present gases directly from the sensor response. It does so by using only a fraction of what conventional learning models need to be fully trained. This has been done by leveraging transfer learning via incorporating the historical data from other sensors toward learning the input-output mapping for a new sensor with only limited available data. This also provides further flexibility as the model can shift from one sensory device to another without requiring large datasets to be fully trained from scratch.
- In a complex environment, could this system work with high performance?
We appreciate the reviewer’s question. We accept that hydrogen monitoring platforms need to be able to work in a harsh and complex environment. As we have proven in another work[6], semiconductor-based commercial sensors used in this work are capable of working in different temperatures and humidity levels. Also, leveraging a strong SAE-TL, this platform can overcome the isolated learning paradigm, reuse a pre-trained model from a different sensor, and apply it to a new sensor and a new mixture of gases.
- Paknahad, M., et al., Characterization of channel coating and dimensions of microfluidic-based gas detectors. Sensors and Actuators B: Chemical, 2017. 241: p. 55-64.
- Paknahad, M., et al., On-Chip Electronic Nose For Wine Tasting: A Digital Microfluidic Approach. IEEE Sensors Journal, 2017. 17(14): p. 4322-4329.
- Barriault, M., et al. A method of accelerated regeneration for a microfluidic gas sensor. in 2017 IEEE SENSORS. 2017.
- Barriault, M., et al., Classification and Regression of Binary Hydrocarbon Mixtures using Single Metal Oxide Semiconductor Sensor With Application to Natural Gas Detection. Sensors and Actuators B: Chemical, 2021. 326: p. 129012.
- Paknahad, M., C. McIntosh, and M. Hoorfar, Selective detection of volatile organic compounds in microfluidic gas detectors based on “like dissolves like”. Scientific Reports, 2019. 9(1): p. 161.
- Paknahad, M., J.S. Bachhal, and M. Hoorfar, Diffusion-based humidity control membrane for microfluidic-based gas detectors. Analytica Chimica Acta, 2018. 1021: p. 103-112.
Round 2
Reviewer 2 Report
The authors have addressed most of my questions.